# Extended Time Window (>6 Hour) Mechanical Thrombectomy; Good Clinical Outcome in the Younger Age Population in Thrombectomy Cases: Relationship between Age and Prognosis

**DOI:** 10.3390/jpm14010002

**Published:** 2023-12-19

**Authors:** Deok Un Gok, So Yeon Kim, Young Chul Na, Jin Mo Cho

**Affiliations:** 1Depatment of Neurosurgery, International St. Mary’s Hospital, Catholic Kwandong University, Incheon 22711, Republic of Korea; 2Department of Neurosurgery, Serim General Hospital, Incheon 21364, Republic of Korea

**Keywords:** stroke, thrombectomy, thrombolytic therapy, extended time window

## Abstract

Background: Mechanical thrombectomy (MT) has become increasingly common in recent years, as studies have shown that it can be an effective treatment for acute ischemic stroke (AIS) caused by large vessel occlusion (LVO). However, the efficacy of MT in the extended time window (6 to 24 h) is still uncertain. Our study aims to evaluate the outcomes of MT in the extended time window for AIS patients. Methods: We reviewed data on AIS patients who received MT beyond six hours of stroke onset from 2015 to 2022. The patients’ occlusions were in the internal carotid artery (ICA), middle cerebral artery (MCA), or posterior circulation. Our evaluation included the modified Rankin scale (mRS) and 90-day mortality rates, as well as complications, such as symptomatic intracranial hemorrhage (sICH). Results: Thirty-one patients were included in this study, with a mean age of 75.6 ± 15.1 years, of whom 54.8% were male. The median NIHSS score at presentation was 17. Successful recanalization (TICI 2b to 3) was achieved in 90.3% of patients and the rate of sICH was 6.4%. No difference was observed between the two age groups. The younger age group (<80 years old) showed a better clinical outcome (mRS 0–2; *p* < 0.05, Fisher’s exact test) compared with the older age group. The overall mortality rate was 6.4%. Conclusion: Our study shows that (MT) can be performed effectively and safely within an extended time window, resulting in satisfactory functional outcomes, particularly in the younger age group.

## 1. Introduction

Acute ischemic stroke (AIS) is one of the leading causes of mortality and disability. Intravenous thrombolysis (IVT) and mechanical thrombectomy (MT) are used for AIS treatment. In general, there is much controversy as to how long MT can be performed after symptom onset. MT is typically performed as soon as possible after a stroke has occurred, ideally within 6 h of symptom onset. It is known that if MT is performed late, more cerebral hemorrhage occurs, so it is better not to perform it. However, recent studies have shown that MT may still be effective up to 24 h after symptom onset in certain patients with large vessel occlusion strokes [1,2,3,4,5,6,7,8,9,10]. The treatment effect is known as time-dependent, at least in the conventional time windows of 4.5 h for IVT and 6h for MT [11,12,13]. MT, in addition to IVT, is known as the standard of care for patients with AIS caused by a large vessel occlusion (LVO) of the anterior circulation within 6 h of stroke onset [14,15]. However, it is not known how much time is adequate for a good result.

The RESILIENT trial (Endovascular Treatment with Stent-Retriever and/or Thromboaspiration vs. Best Medical Therapy in Acute Ischemic Stroke) recently tested the efficacy and feasibility of MT for patients with AIS caused by large vessel occlusion (LVO) within 8 h of symptom onset. The study was terminated early due to clear evidence of the efficacy of MT at the first interim analysis [16].

Furthermore, the DAWN and DEFUSE-3 trials have demonstrated favorable outcomes with the extension of the MT time window up to 24 h [9,17]. However, these studies only enrolled patients with a small initial infarct core. The effect of extending the MT time window remained unproven in all patients. In Korea, due to medical insurance restrictions that only cover the benefits of MT within 8 h, it can be challenging to use MT for cases presenting after this time frame.

At our institution, following the publication of these trials, we adopted a policy of providing MT almost unconditionally for up to 24 h as a principle. We now aim to review its clinical results in terms of efficacy and safety.

## 2. Material and Methods

We conducted a retrospective analysis of data from all consecutive patients who were treated with MT for AIS between 2015 and 2022 if they were treated beyond 6 h of stroke onset. This study was approved by the institutional review board and all patients were treated according to our institution’s protocol for AIS (IRB number: IS23RIMI0012). Eligible patients had an occlusion of the intracranial internal carotid artery (ICA), middle cerebral artery (MCA), or vertebro-basilar artery, and either were ineligible for intravenous alteplase. Patients with a previous modified Rankin scale (mRS) score of 3 or more and those under 18 years old were excluded. MT was performed using a stent-retriever or thrombo-aspiration. We recorded baseline clinical variables, including age, gender, and cardiovascular risk factors (hypertension, diabetes, dyslipidemia, smoking, atrial fibrillation, heart failure, and coronary artery disease), as well as the site of occlusion, onset-to-needle time, and onset-to-recanalization time.

Stroke severity was assessed using the National Institutes of Health Stroke Scale (NIHSS), with a score range of 0 to 42 points. Endovascular treatment success was evaluated through the modified Thrombolysis in Cerebral Infarction scale (mTICI), with a range of 0 to 3, where a score of 2b or 3 indicates more than 50% recanalization of the affected territory. Successful recanalization was defined as an mTICI score of 2b to 3.

The functional outcome was measured using the mRS at 90 days, with a score ranging from 0 (no symptoms) to 6 (death). A score of 2 or less was considered a good clinical outcome in this study [18].

Hemorrhagic transformation was defined as any type of hemorrhage in post-MT imaging, and symptomatic intracranial hemorrhage (sICH) was defined as bleeding greater than 30cc that required surgical treatment. We also evaluated mortality at 90 days.

The patients were divided into two groups: older patients (>80 years) and younger patients. We compared the clinical results and complication rates between the two groups.

SPSS version 24 (IBM Corp., Armonk, NY, USA) was used for statistical analyses. Baseline characteristics were assessed with Fisher exact tests as appropriate using univariate analysis. We performed univariate and multivariate logistic regression analyses (stepwise backward method) to identify independent predictors of good functional outcomes in our sample. The threshold for statistical significance was set as *p* < 0.05.

## 3. Results

Between 2015 and2022, we undertook 321 MTs. Of those,31 (9.7%) were treated in a late time window (6 h to 24 h), defined as the time from symptoms onset to arterial puncture being more than 6 h. All 31 patients with symptom onset within 24 h and major vessel (ICA, MCA, and vertebrobasilar artery) occlusion underwent MT (Figure 1). The mean age was 75.6 ± 15.1 years and 54.8% of patients were male. The median NIHSS score at presentation was 17.

Before the procedure, 5 of the 31 patients received IV alteplase. Vessel occlusions were located in the M1 segment of the MCA in 20 (64.5%), M2 segment of MCA in 4 (12.9%), ICA in 4 (12.9%), and vertebrobasilar artery in 3 patients (9.7%).The performance times (median) were as follows: onset-to-groin time was 12 h 31 min (510 min—1410 min) and onset to recanalization was 13 h 2 min (571 min–1451 min). Successful recanalization (TICI 2b/3) was obtained in 90.3% of patients (*n* = 28). Table 1 summarizes the clinical and imaging outcomes of the included patients. Overall, the 3-month mortality rate was 6.4% (*n* = 2) and Figure 2 shows the distribution of mRS at 90 days (*n* = 31). Overall, 16 (51.6%) patients achieved functional independence (mRS 0–2) at 3 months (Figure 2). ICH occurred in two older patients (6.4%), but there was no need for surgery.

Comparing the two groups according to age, the proportions of patients who could achieve functional independence (mRS 0–2) at 3 months were73.3% (11 out of 15) in the younger population and 31.3% (5 out of 16) in the older population, showing a statistically significantly better prognosis in the younger group (*p* < 0.05, Fisher’s exact test) (Figure 3). Table 2 displays the results of our study compared with those of two recent large clinical trials that evaluated MT for AIS beyond 6 h.

## 4. Discussion

This study provides valuable insights into the efficacy and safety of MT for the treatment of AIS in the extended time window. The results suggest that MT is a viable option for patients presenting with AIS beyond the six-hour time window, with a high rate of successful recanalization and low rates of sICH. Our results showed that MT performed after 6 h also showed good results and the hemorrhagic risk was not so high.

We found that the functional outcomes in the young age group showed better prognosis than those in the two recent large series (mRS 0 to 2 of 73.3% in our series DAWN trial 49%, DEFUSE-3 trial 45%). Given the small sample of our study, we cannot generalize our result to all the patients. Nevertheless, to our knowledge, this is the first case series of an extended time window MT for AIS with LVO in Korea.

Another important finding from this study is the low rate of sICH, which is a potential complication of MT. This suggests that the procedure was performed safely in this group of patients. However, it is worth noting that the sICH rate reported in this study is lower than the rates reported in some other studies of MT in the extended time window [19,20,21]. It is possible that this difference may be due to differences in patient selection or procedural techniques.

The finding that younger age (<80 years old) is associated with better clinical outcomes is interesting and warrants further investigation. This could be due to a variety of factors, such as better overall health or a higher likelihood of having a smaller clot burden. It would be useful to conduct additional studies to explore the reasons behind this age-related difference in outcomes.

It is also important to consider the generalizability of the study results. This study was conducted at a single center, and the patient population may not be representative of all AIS patients who present beyond the six-hour time window. Additionally, this study included only patients with occlusions in the MCA or posterior circulation, and it is unclear whether the results would apply to patients with occlusions in other locations.

Finally, it is worth considering the limitations of this study’s design. As mentioned earlier, this study did not include a control group, and therefore, it is difficult to determine whether the outcomes were solely due to the MT intervention. Additionally, this study did not report on long-term outcomes beyond 90 days, which may be important for understanding the durability of the treatment effect.

This study has several limitations. First, this study included only 31 patients, which may not be representative of the larger population. A larger sample size may be needed to confirm the findings and increase the generalizability of this study.

Second, this study is retrospective in nature, which may introduce bias into data collection and analysis. Prospective studies are needed to confirm the findings of this study.

Third, in the older age group, patients with originally poor mRS were excluded. Although this is not significant, it is thought to have had some influence on the results. We think that this could be a limitation because the older age group often had poor mRS.

Finally, patients who received MT beyond six hours of stroke onset may have been selected based on certain criteria, which may not apply to all AIS patients. This may have limited the generalizability of the findings.

Our results demonstrate superior outcomes compared with both the DEFFUSE 3 and DAWN trials. However, it is important to note a limitation: the sample size was relatively small, raising concerns about the robustness of our findings. In DEFFUSE 3 and DAWN, diffusion MRI or CT scans were conducted on a highly selective group of patients. In contrast, our study included all patients diagnosed from 6 h up to 24 h, introducing the possibility of selection bias.

The inclusive approach in our study suggests that MT can even be considered in cases with diagnosis exceeding 6 h. However, caution is warranted in generalizing these findings at this stage. There is potential for selection bias due to our broad inclusion criteria. Therefore, our results serve as preliminary evidence that MT may be viable in cases beyond 6 h, but further investigation is needed before generalizing these conclusions

The inclusive approach of our study suggests that MT can be considered even in cases exceeding 6 h. However, caution is warranted in generalizing these findings at this stage. There is potential for selection bias due to our broad inclusion criteria. Therefore, our results serve as preliminary evidence that MT may be viable in cases beyond 6 h, but further investigation is needed before generalizing these conclusions.

A follow-up study with a more selective approach utilizing diffusion MRI or perfusion CT would provide valuable insights. In summary, while MT may be considered for cases beyond 6 h, it is premature to draw broad conclusions based on our current findings.

## 5. Conclusions

Our study shows that MT can be performed effectively and safely within an extended time window, resulting in satisfactory functional outcomes without special side effects, particularly in the younger age group. In order to generalize the results of our study, it is thought that a larger randomized study is needed.

## Figures and Tables

**Figure 1 jpm-14-00002-f001:**
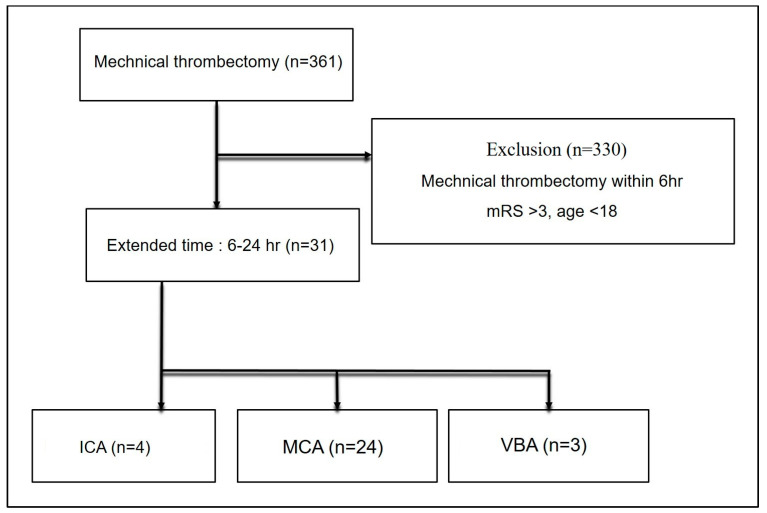
CONSORT flow diagram of disposition of patients enrolled in this study.

**Figure 2 jpm-14-00002-f002:**
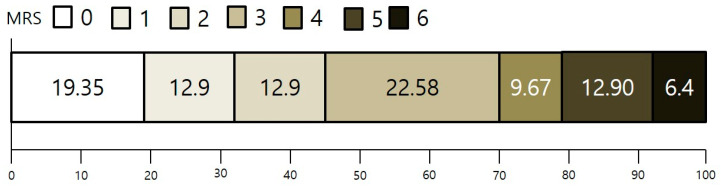
Scores on the modified Rankin scale at 90 days. Modified Rankin scale of patients who underwent MT within 6–24 h. Scores on the modified Rankin scale range from 0 to 6, with 0 indicating no symptoms, 1 indicating no clinically significant disability, 2 indicating slight disability, 3 indicating moderate disability, 4 indicating moderately severe disability, 5 indicating severe disability, and 6 indicating death.

**Figure 3 jpm-14-00002-f003:**
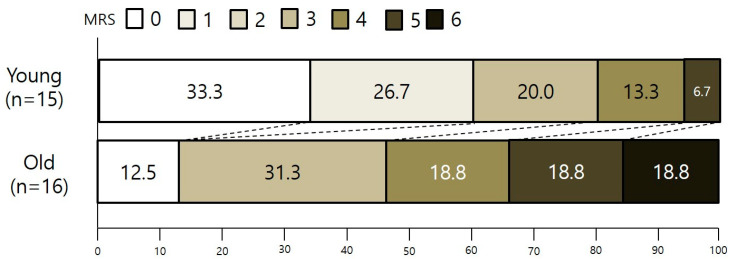
Two groups’ scores on the modified Rankin scale at 90 days. There was a significant difference favoring the younger age group in the overall distribution of scores (unadjusted common odds ratio, 2.42; 95% CI, 1.71 to 4.70; *p* < 0.05).

**Table 1 jpm-14-00002-t001:** Baseline characteristics of the patients and features of thrombectomy.

Variable	All Patients (*n* = 31)
Age (years)	75.66 ± 15.1
Male, *n* (%)	17(54.8)
Hypertension, *n* (%)	24 (77.4)
Smoking, *n*(%)	16 (51.6)
Atrial fibrillation, *n* (%)	5 (16.1)
Intravenous thrombolysis	5 (16.1)
NIHSS, median	17 [12–20]
Occlusion site, *n* (%)	
MCA, M1	20 (64.5)
MCA M2	4 (12.9)
ICA	4(12.9)
Vertebrobasial artery	3 (9.7)
Onset to groin time	12 h 31 min (510 min–1410 min)
Onset to recanalizationtime	13 h 2 min (571 min–1451 min)

**Table 2 jpm-14-00002-t002:** The outcomes of the present study compared with two large trials on mechanical thrombectomy treatment for AIS beyond 6 h.

Trial, Year	Sample (*n*)	Intravenous Thrombolysis *n* (%)	Recanalization (mTICI = 2b/3) *n* (%)	3 Months mRS ≤ 2 *n* (%)	sICH	3 Months Mortality *n* (%)
DAWN, 2017 [9]	107	5 (5)	90 (84)	20 (19)	6 (6)	20 (19)
DEFUSE 3, 2018 [17]	92	10 (11)	69 (76)	13 (14)	6 (7)	13 (14)
Present study, 2023	31	5 (16.1)	28 (90.3)	16 (51.6)	2 (6.4)	3 (9.7)
Present study (younger patients), 2023	15	2(13)	15 (93)	12(80)	0 (0)	0 (0)

## Data Availability

The data presented in this study are available on request from the corresponding author.

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
