# Peer review of "Extended Time Window (>6 Hour) Mechanical Thrombectomy; Good Clinical Outcome in the Younger Age Population in Thrombectomy Cases: Relationship between Age and Prognosis"

_jpm, 2023, doi:10.3390/jpm14010002_

Round 1

Reviewer 1 Report

Comments and Suggestions for Authors

I have read with interest your work on the use of mechanical thrombectomy during an extended time window.

This experience reinforces and complements the evidence emerging from the DAWN and DEFUSE-3 trials.

Please find below some minor changes that could improve the communication of your work:

- in line 62 it is stated that "Because they arrived late, none of the patients enrolled here were able to receive IVT", while in line 98 it is stated that "Before the procedure, 5 of 31 patients received IV alteplase". Please correct this inconsistency.

- The group of patients with a younger age cannot properly be defined as a group of patients with a "young age". Please correct the definition of the two groups with a more appropriate one, e.g. "younger patients" and "older patients".

- Please insert in table 1 the number of patients who met the inclusion criteria for the DAWN and DEFUSE-3 trials, respectively.

- Please insert a row in table 2 about the outcomes of the entire population studied in your retrospective cohort study.

- Please report the outcomes relating to the entire population in the main text: recanalization, 3 months mRSz=2, sICH, 3 months mortality.

- Please correct the misspelling of recanalization in table 2.

Author Response

Reviewer 1

- in line 62 it is stated that "Because they arrived late, none of the patients enrolled here were able to receive IVT", while in line 98 it is stated that "Before the procedure, 5 of 31 patients received IV alteplase". Please correct this inconsistency.

Thanks for the good point. Actually, I did not include the IVT patient group at first because there was no indication, but I included it because I thought it would be right to include the actual treatment performed. That phrase has been deleted.

- The group of patients with a younger age cannot properly be defined as a group of patients with a "young age". Please correct the definition of the two groups with a more appropriate one, e.g. "younger patients" and "older patients".

Thanks for the good point. We made changes based on the reviewer's comments and marked the revised parts.

- Please insert in table 1 the number of patients who met the inclusion criteria for the DAWN and DEFUSE-3 trials, respectively.

I don't understand this part very well. Tab 1 is about this study. What does it mean to include people who are part of the DAWN and Deffuse-3 trial ? I don't understand, so I'd like to ask again about this.

- Please insert a row in table 2 about the outcomes of the entire population studied in your retrospective cohort study.

It was added based on the reviewer's opinion.

- Please report the outcomes relating to the entire population in the main text: recanalization, 3 months mRSz=2, sICH, 3 months mortality.

It was added based on the reviewer's opinion.

- Please correct the misspelling of recanalization in table 2.

I’m sorry. There was a mistake. It was revised according to the reviewer's opinion.

Reviewer 2 Report

Comments and Suggestions for Authors

What kind of procedure techniques do the authors think accounted for the low rates of sICH in their study compared to others?

What are the rate of minor bleeding that did not require surgical treatment?

Please describe the anti-platelet and anticoagulation strategy for the immediate post op and near short term for these patients

Comments on the Quality of English Language

Minor changes - 

1st line of results section needs to be re-phrased

Need spelling and grammar check - words don't have space in between them

Author Response

Reviewer 2

What kind of procedure techniques do the authors think accounted for the low rates of sICH in their study compared to others?

I think the biggest reason is that the sample size is small. This was stated as a limitation of our study.

What are the rate of minor bleeding that did not require surgical treatment?

ICH occurred in 6.4% (n=2). I think it will increase further if the sample size becomes larger.

Please describe the anti-platelet and anticoagulation strategy for the immediate post op and near short term for these patients

As a rule, I used dual anti-platelet for at least 2 years and did not stop using it if possible. I didn't add it to the main text because it seems unrelated to the topic of the paper. Should I add it?

1st line of results section needs to be re-phrased

There was an error in the year, so it has been corrected.

Need spelling and grammar check - words don't have space in between them

We reviewed and revised it in detail.